# Unraveling the drivers of leptospirosis risk in Thailand using machine learning

Pikkanet Suttirat[1,2], Sudarat Chadsuthi[3], Charin Modchang [1,2,4]*, Joacim Rocklöv[5,6,7]*

**1** Biophysics Group, Department of Physics, Faculty of Science, Mahidol University, Bangkok, Thailand, **2** Center for Disease Modeling, Faculty of Science, Mahidol University, Bangkok, Thailand, **3** Department of Physics, Faculty of Science, Naresuan University, Phitsanulok, Thailand, **4** Centre of Excellence in Mathematics, MHESI, Bangkok, Thailand, **5** Department of Epidemiology and Global Health, Umeå University, Umeå, Sweden, **6** Heidelberg Institute of Global Health, Heidelberg University, Heidelberg, Germany, **7** Interdisciplinary Center of Scientific Computing, Heidelberg University, Heidelberg, Germany

\* charin.mod@mahidol.edu (CM); joacim.rocklov@umu.se (JR)

## Abstract

Leptospirosis poses a significant public health challenge in Thailand, driven by a complex mix of environmental and socioeconomic factors. This study develops an XGBoost machine learning model to predict leptospirosis outbreak risk at the provincial level in Thailand, integrating climatic, socioeconomic, and agricultural features. Using national surveillance data from 2007-2022, the model was trained to classify provinces as high or low risk based on the median incidence rate. The model's predictive performance was validated for the years 2018–2022, spanning pre-COVID-19, COVID-19, and post-COVID-19 periods. SHapley Additive exPlanation (SHAP) analysis was employed to identify key predictive factors. The optimized XGBoost model achieved high predictive accuracy for the pre-pandemic (AUC = 0.937 with 95% CI: 0.878 − 0.976) and post-pandemic (AUC = 0.951 with 95% CI: 0.861 − 0.999) testing periods. SHAP analysis revealed rice production factors, household size, and specific climatic variables as the strongest predictors of leptospirosis risk. However, model performance declined during the COVID-19 pandemic (2020–2021), suggesting surveillance disruption and potential underreporting. This study demonstrates the utility of machine learning for predicting leptospirosis risk in Thailand and highlights the complex interplay of environmental and socioeconomic factors in driving outbreaks. The adaptable modeling framework provides a foundation for developing early warning systems and targeted interventions to reduce the burden of this neglected tropical disease.

## Author summary

Leptospirosis, a disease caused by *Leptospira* bacteria, poses a significant public health challenge in Thailand. The bacteria thrive in contaminated

**Data availability statement:** The code and data underlying the results presented in the study are available from https://doi.org/10.5281/zenodo.15161968.

**Funding:** PS is supported by the Development and Promotion of Science and Technology Talents Project (DPST) of Thailand. SC was supported by National Research (NU) and National Science, Research and Innovation Fund (NSRF) (Grant NO. R2568B013). CM was funded by The Rockefeller Foundation under the Strengthening the Early Warning and Outbreak Detection Systems through Nation-wide Event-based and Syndromic Surveillance (STONES) project (Grant No. 2021PPI005). The funders had no role in study design, data collection and analysis, decision to publish, or preparation of the manuscript.

**Competing interests:** The authors have declared that no competing interests exist.

environments, particularly those associated with rice farming. In this study, we developed a machine learning model to predict the risk of leptospirosis outbreaks in Thailand based on climatic, socioeconomic, and agricultural factors. Our analysis revealed that rice production practices, household size, and specific climatic variables were the strongest predictors of leptospirosis risk. We also observed a reduction in model performance during the COVID-19 pandemic, suggesting surveillance disruptions and potential underreporting. These findings highlight and explain the complex interplay of environmental and socioeconomic factors in driving leptospirosis outbreaks. Our adaptable modeling framework provides a foundation for developing early warning systems and targeted interventions to reduce the burden of this often-overlooked tropical disease. Better understanding the factors that contribute to leptospirosis risk can guide responses to protecting vulnerable populations and improving public health outcomes in Thailand and beyond in times of socio-environmental changes.

## Introduction

Leptospirosis, caused by the spirochete bacterium *Leptospira*, represents a significant global public health challenge, particularly in tropical and subtropical regions. The bacteria can persist in soil and water for months under suitable environmental conditions, with human infection occurring primarily through contact with contaminated environments that provide ideal conditions for *Leptospira* growth [1,2]. The disease burden is substantial, with estimates from 2015 indicating over 1 million cases and 58,000 deaths globally each year [3]. The challenge of managing leptospirosis is compounded by its complex clinical presentation, as initial symptoms often mirror those of other diseases, such as dengue fever and COVID-19, making early diagnosis particularly challenging [4,5]. The infection spectrum ranges from asymptomatic cases to severe acute infections that can prove fatal [6,7].

The epidemiology of leptospirosis is intrinsically linked to social, environmental and climatic interactions, with distinct patterns emerging in different geographical contexts. In Thailand, studies have identified several key determinants of leptospirosis outbreaks, including primary rice-planted areas, local elevation, and population density [8]. Research has also highlighted the significance of topographic slope variance, precipitation, maximum temperature, and flood-prone areas in influencing the spatial distribution of the disease [9]. Similar associations between climatic factors and leptospirosis outbreaks have been documented globally [10–13], with flooding consistently emerging as a critical trigger for outbreaks across multiple countries [10,14,15]. Beyond environmental factors, socioeconomic indicators such as gross domestic product and population density have been shown to influence human leptospirosis infection rates [11].

Traditionally, researchers have employed two main approaches to investigate the impact of climatic and socioeconomic factors on leptospirosis risk. The first approach utilizes mechanistic transmission models, which have proven successful in predicting

infectious disease transmission patterns [16–19]. However, these models often face challenges in parameter estimation due to their inherent complexity and underlying assumptions. The second approach employs statistical methods to analyze spatial and temporal surveillance data, requiring extensive data collection and preprocessing, and careful consideration of variable interactions [20,21].

Recently, machine learning models have emerged as a promising alternative method for disease modeling and outbreak prediction. In particular, "eXtreme Gradient Boosting" (XGBoost) has gained prominence across various scientific fields due to its accuracy and ease of application [22–25]. XGBoost offers the advantage of handling large sets of input variables with minimal preprocessing requirements. While machine learning algorithms like XGBoost can be complex and initially challenging to interpret due to their "black-box" nature, recent developments in eXplainable Artificial Intelligence (XAI) have made it possible to better understand and interpret these models [26,27]. Especially, Shapely values have emerged as a tool for interpretation of complex features combinations in machine learning models, borrowing strength from game theory by considering predictions as a "game" outcome and features as "players" [28].

The potential of machine learning in epidemiological research has been demonstrated in recent studies, such as the investigation of eco-climatic factors in West Nile virus outbreaks in Europe [29]. In Thailand, while machine learning has been applied to predict leptospirosis prevalence using environmental data [9], there remains a critical need for a comprehensive framework that integrates historical data, climatic factors, and socioeconomic influences to predict and explain leptospirosis outbreak drivers effectively.

In this study, we develop and evaluate an XGBoost model that combines climatic, socioeconomic, and rice production factors to predict leptospirosis outbreak risk in Thailand. Using surveillance data from 2007 to 2017, we train the model to distinguish between low-risk and high-risk provinces based on median incidence rates. We then validate the model's predictive capabilities for the years 2018–2022 on testing data withheld from model training, encompassing pre-COVID-19, COVID-19, and post-COVID-19 periods. To enhance model interpretability, we employ the SHapley Additive exPlanation (SHAP) method to describe the influence of various predictive factors on the outbreak risk. Additionally, we analyze how the COVID-19 pandemic influenced our model's predictions by comparing performance across different temporal periods.

## Materials and methods

### Data collection

Human leptospirosis cases were obtained from national disease surveillance (report 506) provided by the Bureau of Epidemiology, Department of Disease Control, the Ministry of Public Health, Thailand [30]. Positive cases are defined as suspected cases reported from hospitals based on clinical diagnostic and risk assessment. In this research, yearly reported cases between 2007 and 2022 at the provincial level were analyzed. Reported cases from the Buengkarn province were considered included in the Nong Khai province during the time they were administratively combined (2007–2011), i.e., the Buengkarn province still was not yet established. All features were calculated according to the province's border at the time.

Climatic variables were obtained from the TeraClimate dataset [31], including the average monthly value of precipitation, minimum temperature, maximum temperature, soil moisture, and vapor pressure. The data are provided in 4 km$^2$ gridded format and are aggregated to the provincial level. The annual average of each factor was calculated from the average monthly value. The climatic variables of Nong Khai province during 2007 – 2011 were calculated from the area-weighted average between Nong Khai and Buengkarn values. A set of 19 bioclimatic features (bio1 -bio19) was then derived from monthly temperature and precipitation data using the R package '*dismo'* at the provincial level [32,33]. Bioclimatic factors represent annual trends, seasonality, and fluctuating environmental factors throughout the year.

We included primary (in-season) and secondary (off-season) agricultural rice production statistics obtained from a dataset provided by the Office of Agricultural Economics via the Open Government Data platform (https://data.go.th/en/

dataset/oae0001). Two rice-related factors were included in the potential features, namely, rice cultivated area and rice production yield. Economic factors were also used as potential features. In addition, gross regional and provincial product (GPP) was used as an indicator for the provincial economic index. The inflation-adjusted GPP using the Laspeyres Index is obtained from the office of the National Economic and Society Development Council (https://www.nesdc.go.th). Population statistics and household size were obtained from the Official registration statistics systems of the Bureau of Registration Administration (https://stat.bora.dopa.go.th). The mean income per household was acquired from the National Statistical Office of Thailand, where the household incomes were reported every two years (https://www.nso.go.th). We interpolated the missing year income value by linear interpolation. The full description and source of the variables are presented in Table 1.

## Data processing and model selection

We split the available data into a training set (from 2007 to 2017, 11 years in total) and a testing set (from 2018 to 2022, 5 years in total). We calculated the incidence rate per 100,000 population of a province as representative of leptospirosis risk within the province. The training set was used to construct the model and determine the optimized model parameters. Later, the final optimized model was applied to predict the test years outbreak risk, i.e., 2018–2022, to assess the model performance on the unseen data, including the pre-, during, and post-COVID-19 pandemic period.

For our classification model, we calculated the annual incidence rate per 100,000 population for each province as an indicator of leptospirosis outbreak level. We selected a risk classification threshold of 2.43 cases per 100,000 population, corresponding to the nationwide median incidence rate during the training period (2007–2017). This threshold effectively distinguished between provinces with elevated outbreak risk and those with baseline endemic levels, dividing the data into high- and low-risk classes.

To solve the classification task of the leptospirosis risk, the supervised machine learning model was utilized to learn complex feature interactions in the data during the training set to determine the class of each province based on input features. We chose XGBoost as the algorithm for solving this classification task. It utilizes decision trees capable of solving classification tasks based on the combination of the gradient boosting tree algorithm and approximate tree learning to improve computation efficiency [22].

The complicated nature of XGBoost feature combinations is often hard to interpret, especially at the single observation level of prediction. Therefore, we adopted SHapley Additive exPlanation (SHAP) for interpreting the predictors features influence from the gradient-boosted decision tree model trained on leptospirosis data. SHAP is a game-theory-based framework on explainable AI principles that aims to explain the output of any machine learning model at the single-prediction level [34]. SHAP calculated values based on the comparison between model prediction with and without a feature for all possible combinations of features at every single observation. Then, features were ranked according to their SHAP value influence according to how it impacted model predictions.

## Cross-validation and hyperparameter tuning

We chose a 5-fold cross-validation strategy to ensure model generalization ability on the data while not excessively increasing the computational cost. The data was partitioned into five folds (subsets) of equal size while maintaining an identical ratio between positive and negative classes. Then, the model iteratively selected one of the subsets to be left out of the training process. The left-out data was used to evaluate model performance in that fold. The overall performance of each machine learning model was evaluated from the average score between each iteration.

Hyperparameter tuning is an important aspect of machine learning as machine learning algorithms can be sensitive to their structural model parameters. Our model hyperparameters determined the *number of trees*, the *learning rate, the maximum depth, subsampling, column sample by tree, minimum child weight, gamma, alpha, and lambda.* They were optimized during the cross-validation process, searching for the best combination of hyperparameters utilizing a Python

**Table 1. Variables/Features used to identify leptospirosis risk from XGBoost and their description.**

| Variable/ Feature | Feature description | Reference |
|---|---|---|
| **Climate** | | https://www.climatologylab.org/terraclimate.html |
| ppt | Annual average of precipitation (mm) | |
| tmin | Annual average of minimum temperature (°C) | |
| tmax | Annual average of maximum temperature (°C) | |
| vap | Annual average of vapor pressure (kPa) | |
| soil | Annual average of soil moisture (mm) | |
| **Bioclimatic** | **Temperature-related bioclimatic feature bio1-bio11** | https://rdrr.io/cran/dismo/man/biovars.html [33] |
| bio1 | Mean annual temperature | |
| bio2 | Mean diurnal range (mean of max temp - min temp) | |
| bio3 | Isothermality (bio2/bio7) (* 100) | |
| bio4 | Temperature seasonality (standard deviation *100) | |
| bio5 | Max temperature of the warmest month | |
| bio6 | Min temperature of the coldest month | |
| bio7 | Temperature annual range (bio5-bio6) | |
| bio8 | Mean temperature of the wettest quarter | |
| bio9 | Mean temperature of driest quarter | |
| bio10 | Mean temperature of warmest quarter | |
| bio11 | Mean temperature of coldest quarter | |
| | **Precipitation-related bioclimatic feature bio12-bio19** | |
| bio12 | Total (annual) precipitation | |
| bio13 | Precipitation of wettest month | |
| bio14 | Precipitation of driest month | |
| bio15 | Precipitation seasonality (coefficient of variation) | |
| bio16 | Precipitation of wettest quarter | |
| bio17 | Precipitation of driest quarter | |
| bio18 | Precipitation of warmest quarter | |
| bio19 | Precipitation of coldest quarter | |
| **Rice Production** | | https://data.go.th/dataset/oae0001 |
| MRiceArea | Primary rice cultivated area ($km^2$) | |
| MRiceYield | Primary rice yield (metric ton/$km^2$) | |
| SRiceArea | Second rice cultivated area ($km^2$) | |
| SRiceYield | Second rice yield (metric ton/$km^2$) | |
| **Socioeconomic** | | |
| GPP | Gross provincial product (million baht) | https://www.nesdc.go.th/en/info/gross-regional-and-provincial-product-gpp/ |
| HouseholdSize | Mean household size (person) | https://stat.bora.dopa.go.th/new_stat/webPage/statByYear.php |

*(Continued)*

**Table 1.** (Continued)

| Variable/ Feature | Feature description | Reference |
|---|---|---|
| income | Mean household income (baht) | https://www.nso.go.th/nsoweb/nso/statistics_and_indicators?impt_branch=309 |
| **Predicted feature** | | |
| Predicted Feature/ Risk (binary) | Provine with higher (1) and lower (0) than median incidence rate (reported cases per 100,000 population) | |

package, *Optuna*, and a Bayesian optimization method named Tree-Structured Parzen Estimator (TPE) sampling. The latter utilizes a performance-aware searching space in which the parameter space is searched more often if the historical parameters set give a higher objectives function. TPE has benefits over random searching because the algorithm balances between searching time and exploration of parameter space [35].

In our hyperparameter tuning process, a metric *logloss* of the training set was set as a determiner score during the cross-validation process. *Logloss* measures the difference between the data and prediction class (0 or 1). XGBoost additionally introduced the early-stopping mechanism to further prevent overfitting the boosted tree to the training set. In this work, we set this early stopping round to 30 (see also S1 Text). This proposed framework was implemented using *Python* 3.10 with standard packages such as *pandas*, *numpy*, *scikit-learn,* and *XGBoost*.

## Performance metrics

We chose a metric called Area Under the receiver operating characteristic Curve (AUC) as a metric for model performance. AUC is designed to measure model performance regardless of classification threshold. It shows the tradeoff between a true positive rate (sensitivity) and a false positive rate (1-specificity) upon the various classification thresholds. Other threshold-dependent evaluation metrics at the various thresholds to show the overall performance of the model, e.g., accuracy represents the overall model performance in both positive and negative classes (Eq. 1), precision represents the fraction of positive prediction that is correctly predicted (Eq. 2), sensitivity represent fraction of positive class (high-risk class) that model can capture (Eq. 3) and F1-score represents a tradeoff between precision and sensitivity by calculate harmonic mean between two value (Eq. 3). For each threshold, the predicted probability output of the XGBoost model is compared to the threshold values. If the predicted probability output is higher than a threshold, the predicted result is high risk (1). Otherwise, the predicted class is low risk (0).

$$Accuracy = \frac{TP + TN}{TP + TN + FP + FN} \tag{1}$$

$$Precision = \frac{TP}{TP + FP} \tag{2}$$

$$Sensitivity = \frac{TP}{TP + FN} \tag{3}$$

$$F1 - score = \frac{2\ TP}{2\ TP + FP + FN} \tag{4}$$

where *TP* is the number of True Positive predictions, *TN* is the number of True Negative predictions, *FP* is the number of False Positive predictions, and *FN* is the number of False Negative predictions.

## Results

### Spatiotemporal patterns of leptospirosis in Thailand

From 2007 to 2022, Thailand reported 51,299 leptospirosis cases. The northeastern region experienced the highest regional incidence rate of 17.56 cases per 100,000 population (3,774 cases among 21,495,825 people) in 2009. Between 2012 and 2019, both northeastern and southern regions maintained consistently high incidence rates. A notable shift occurred from 2019 onwards, when the northeastern region's incidence rate decreased to levels comparable with other regions, while the southern region continued to exhibit high rates (Fig 1a).

The provincial-level analysis revealed persistent high-risk areas, particularly in the northeastern and southern regions, as shown by the median incidence rates (Fig 1b). We established a risk classification threshold of 2.43 cases per 100,000 population, derived from the nationwide median during the training period (2007–2017). This threshold was used to categorize provinces into high and low-risk classes (Fig 1c). The machine learning model was then developed to analyze the relationships among multiple risk factors, including climatic, bioclimatic, socioeconomic, and rice production variables, noting significant multicollinearity among certain factors (S2 Fig).

### Model performance and key risk factors

We evaluated the XGBoost model's performance using 5-fold cross-validation with logloss optimization for hyperparameter tuning. The model demonstrated strong predictive capability, albeit with varying accuracy across different provinces (Fig 2). Several provinces, including Bangkok, Phichit, and Ranong, showed perfect prediction patterns, while others such as Kalasin, Chumphon, and Chiang Mai exhibited mixed prediction accuracy. The model successfully identified regional patterns, accurately capturing low risk in the central region and high risk in the northeastern and southern regions. The optimized XGBoost model achieved an AUC of 0.989 (95% CI: 0.982 – 0.995) for the training set (2007–2017) and 0.937 (95% CI: 0.878 – 0.976) for the pre-COVID-19 period (2018–2019) (Fig 3a). Additional performance metrics, including accuracy, precision, sensitivity, and F1-score, confirmed the model's strong discriminative ability between high and low-risk provinces during 2007–2019 (S1 Fig).

Temporal stratification of model predictions revealed nuanced performance patterns. The model accurately captured leptospirosis risk dynamics prior to 2020, but systematically overpredicted risk during 2020–2021 (Fig 3a). This overprediction pattern suggests that COVID-19-related interventions, including mobility restrictions, behavioral modifications, and altered healthcare-seeking patterns, may have suppressed leptospirosis transmission through mechanisms independent of the environmental and socioeconomic drivers incorporated in our model.

SHAP analysis of the 2007–2019 period revealed that rice production factors were the strongest predictors of leptospirosis risk (Fig 3b). Secondary and primary rice yields ranked as the first and second most influential factors, respectively, while their corresponding cultivated areas ranked fourth and fifth. Household size emerged as the third most important factor, showing a positive correlation with leptospirosis risk. Among climatic factors, vapor pressure (6th), maximum temperature (8th), and precipitation (10th) demonstrated moderate influence. The analysis also identified gross provincial product (GPP) and precipitation during the driest month (bio14) as significant contributors to risk prediction.

### Leptospirosis during and after the COVID-19 pandemic

The model's predictive performance varied significantly during the pandemic period. Annual AUC calculations revealed a notable decline in 2021, representing the lowest performance of the study period and suggesting the influence of unmeasured pandemic-related factors such as mobility restrictions, behavioral modifications, and altered healthcare-seeking patterns (Figs 4a and S3). This pattern was consistent across all performance metrics (Fig 4b). However, model performance recovered in 2022, returning to pre-pandemic levels with an AUC of 0.951 (95% CI: 0.861-0.999).

PLOS Neglected Tropical Diseases

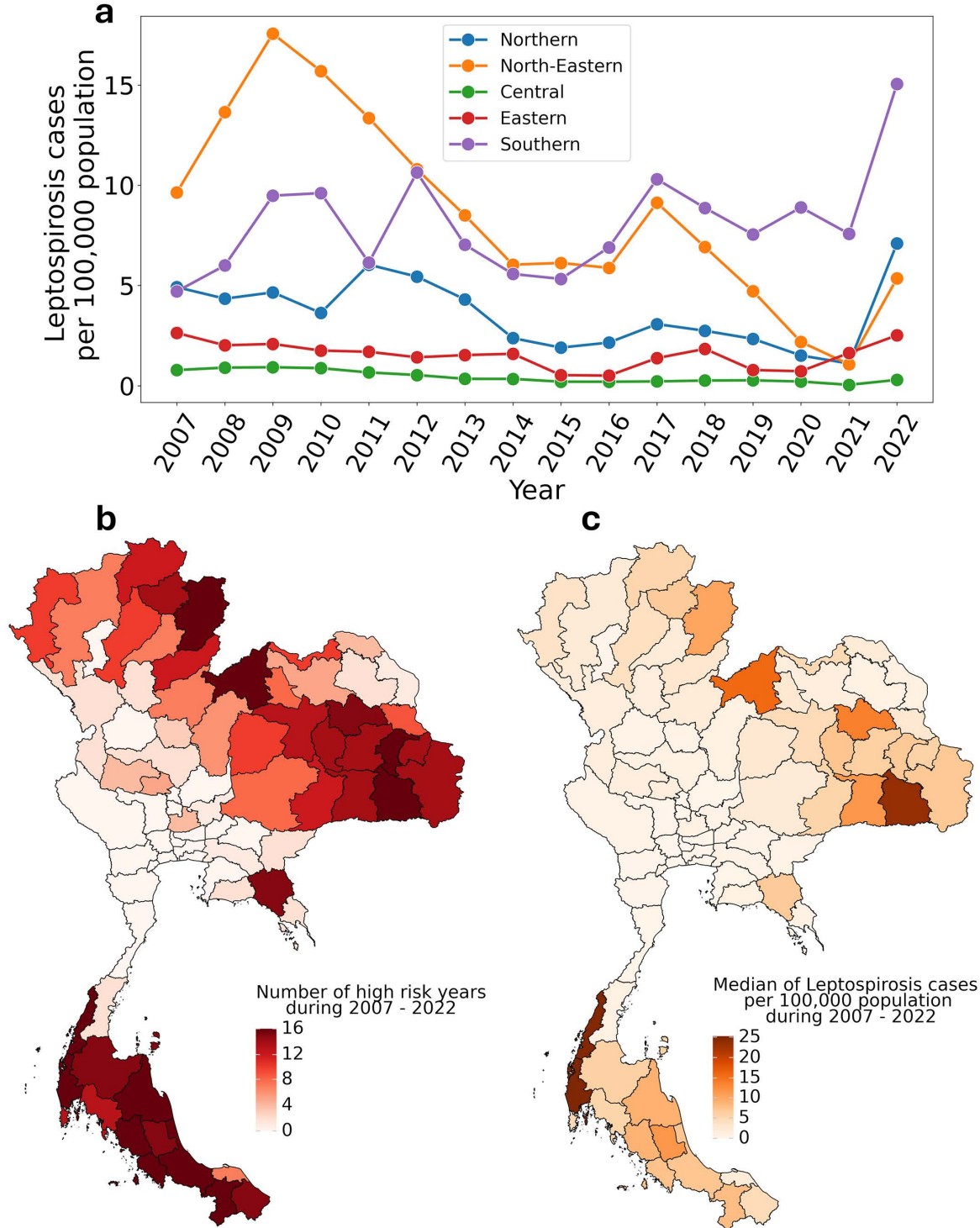

**Fig 1. Spatiotemporal distribution of leptospirosis in Thailand (2007–2022). (a)** Regional trends in annual incidence rates per 100,000 population across Thailand's five major regions, showing distinct temporal patterns. **(b)** Geographic distribution of median incidence rates by province, with darker orange indicating higher rates of reported cases (cases per 100,000 population). **(c)** Frequency of high-risk years per province, where high risk is defined by an incidence rate exceeding the national median of 2.43 cases per 100,000 population (calculated from the 2007–2017 training period). The map plots were created using the plotnine package in Python using a shapefile obtained from The Humanitarian Data Exchange (HDX) (https://data.humdata.org/dataset/cod-ab-tha).

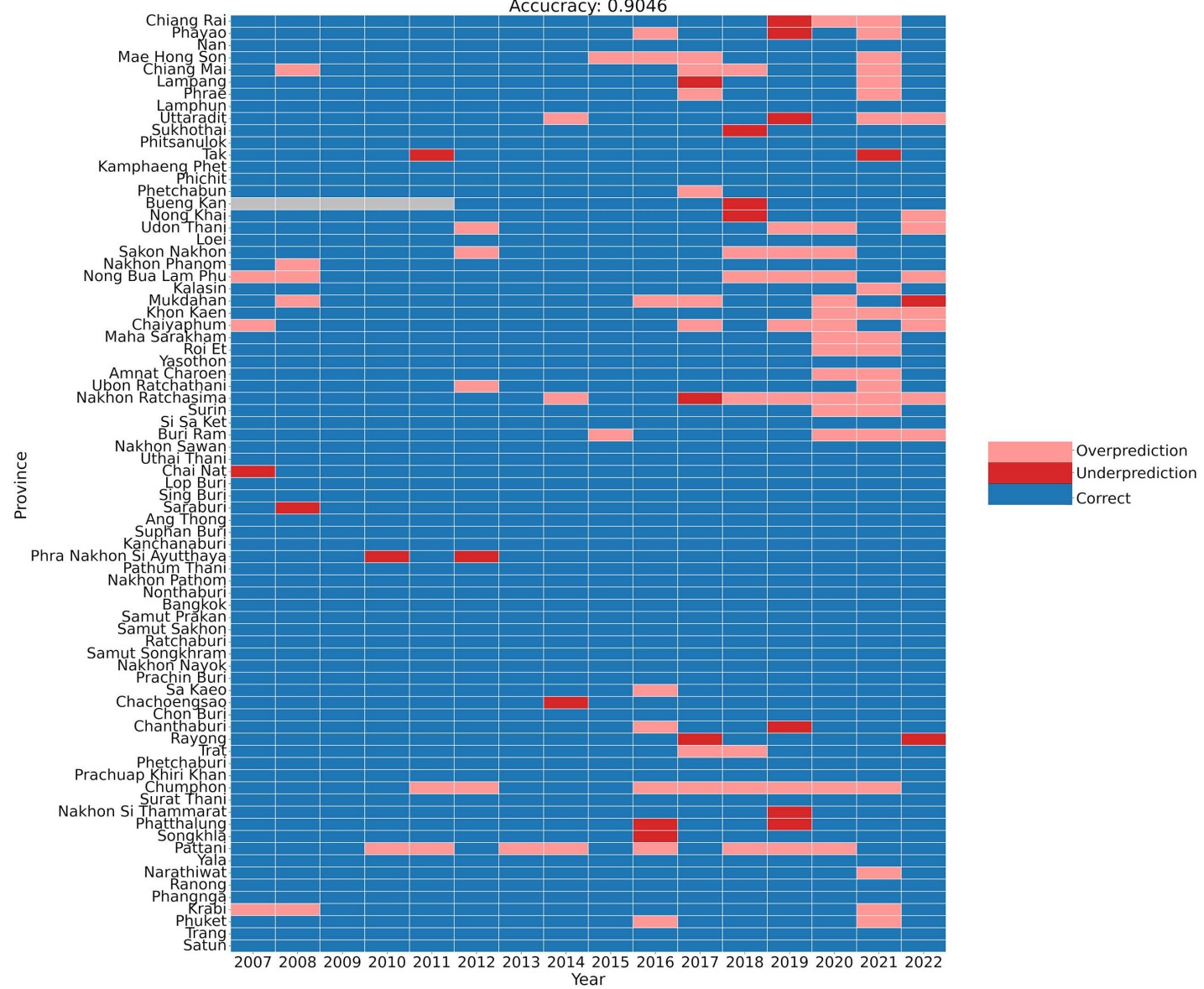

**Fig 2. Temporal and spatial evaluation of XGBoost model predictions across Thai provinces (2007–2022).** The heatmap visualizes prediction accuracy across training (2007–2017) and testing (2018–2022) periods, with colors indicating prediction outcomes: correct predictions (blue), false positives/overprediction (pink, high-risk predicted for low-risk years), and false negatives/underprediction (red, low-risk predicted for high-risk years). Provinces are organized by geographical region and arranged north to south based on latitude. Gray cells indicate missing data from Buengkarn province prior to its establishment in 2012. Overall model accuracy for the entire period (2007–2022) is 0.9046.

## Discussion

### Employing XGBoost to unravel complex risk factors in leptospirosis prediction

Understanding the complex interactions between climatic and socioeconomic factors is essential for predicting leptospirosis transmission. Our study employed XGBoost, a state-of-the-art machine learning algorithm, to analyze leptospirosis risk based on climatic and socioeconomic factors in Thailand. XGBoost's selection was motivated by its demonstrated

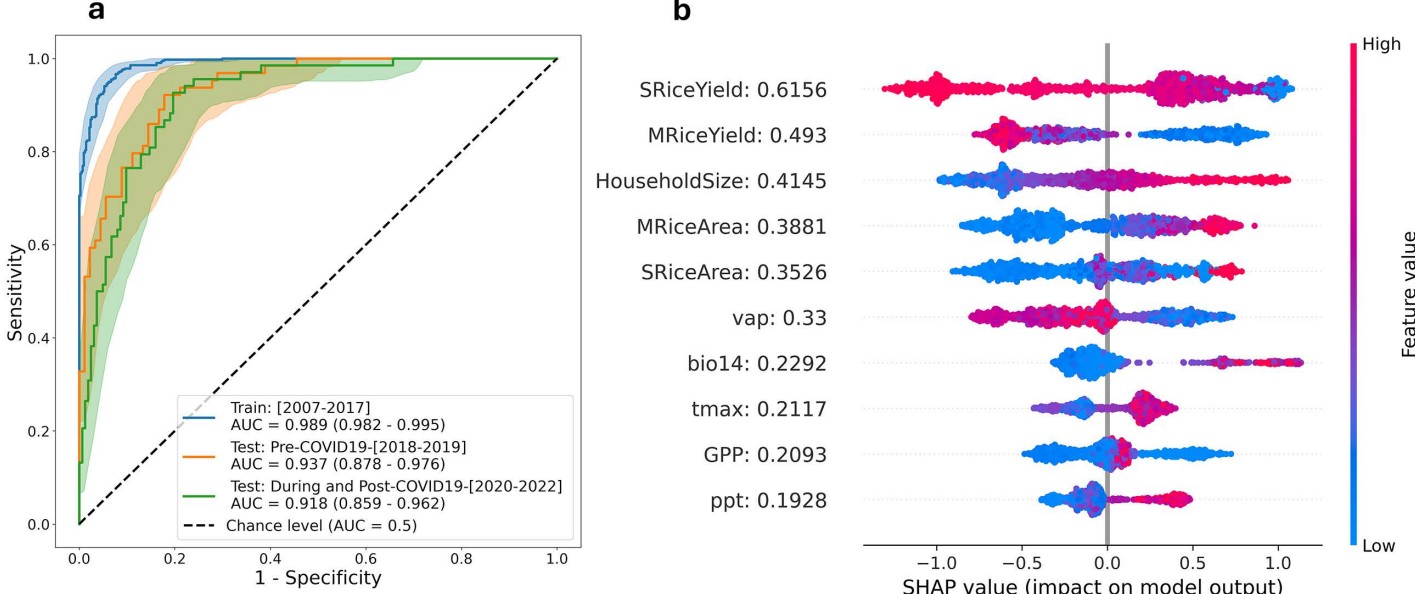

**Fig 3. XGBoost model performance metrics and SHAP-based feature importance analysis. (a)** Receiver operating characteristic (ROC) curves showing model discrimination ability across different datasets, with corresponding area under the curve (AUC) values. Results are derived from the optimal hyperparameters identified through 5-fold cross-validation. The diagonal line represents random classification (AUC = 0.5), while AUC = 1.0 indicates perfect prediction. The shaded area represents 95% confidence intervals (95% CIs) based on 1,000 bootstraps [36]. 95% CIs of AUC were given in parentheses. **(b)** SHAP (SHapley Additive exPlanations) analysis of the top 10 predictive features, ranked by mean absolute SHAP values (2007–2022). Individual observations are represented as dots, with color intensity indicating feature magnitude (red = high, blue = low). SHAP values on the x-axis show each feature's contribution to the predicted probability, centered at log odds = 0 (corresponding to a predicted probability of 0.5).

superiority in handling tabular data with complex, non-linear interactions [22], consistently outperforming traditional algorithms like logistic regression, random forest, and support vector machines in outbreak prediction [29,37,38]. While traditional machine learning approaches struggle with highly correlated features without extensive feature elimination, XGBoost's boosting algorithm effectively manages complex feature interactions through iterative ensemble learning. This process sequentially builds weak learners, minimizing error at each step until convergence, an approach that has proven effective for feature selection in previous studies [39–41]. In fact, our preliminary analysis comparing XGBoost, logistic regression, support vector machine, and random forest confirmed that XGBoost outperformed all other algorithms during both pre-COVID-19 and post-COVID-19 testing periods (S1 Table).

## Impact of rice farming practices

Based on SHAP analysis, our machine learning algorithms captured the importance of the rice yield and rice cultivated area as the top contributing factors to the leptospirosis risk (Fig 3b). This indicated that leptospirosis is the disease in rural areas in Thailand where the rice farming profession is still dominant as the main profession in some provinces. The negative impact of both primary and secondary rice yield indicated that traditional rice farming practices are more prone to leptospirosis infection than more mechanized processes. Higher rice yield in the central region of Thailand might also be attributed to better irrigation availability despite the lower total cultivated area than the northeastern region [42]. These findings align with previous research documenting the relationship between rice farming and leptospirosis outbreaks in Thailand [8,43] and other countries [44–46].

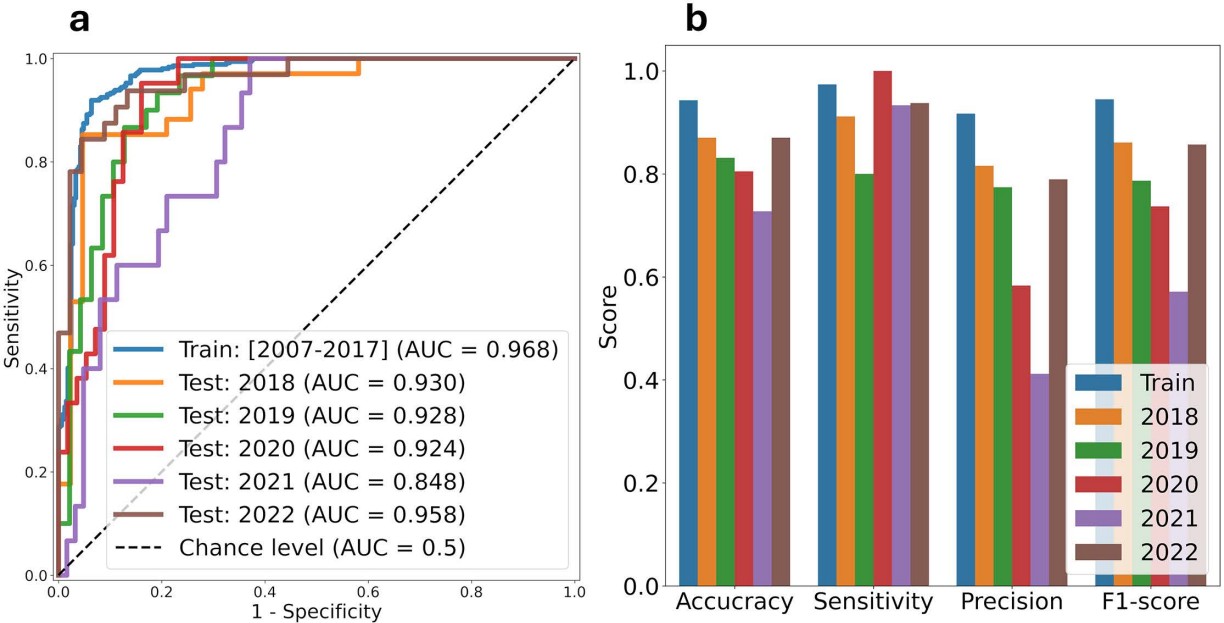

**Fig 4. Model performance evaluation across pre-pandemic, pandemic, and post-pandemic periods. (a)** Receiver operating characteristic (ROC) curves and corresponding area under the curve (AUC) values comparing model discrimination ability across three distinct periods: pre-pandemic (2007-2018), pandemic (2019-2021), and post-pandemic (2022). The diagonal reference line indicates random classification (AUC = 0.5), with perfect classification at AUC = 1.0. To avoid curve cluttering, 95% confidence intervals of AUC are shown in S4 Fig. **(b)** Annual performance metrics using a binary classification threshold of 0.5, including accuracy, precision, sensitivity, and F1-score. Each bar represents a year's performance, demonstrating the temporal evolution of model reliability and highlighting the pandemic's impact on predictive accuracy.

## Socioeconomic disparities and leptospirosis vulnerability

In addition to the role of rice farming, our analysis identified household size as a crucial predictor of leptospirosis risk, underscoring the disease's socioeconomic dimensions. The results indicate that geography-specific factors, rather than year-to-year climatic variations, primarily drive outbreak patterns in Thailand. The disease burden disproportionately affects rural areas with larger household sizes, contrasting with urban areas characterized by smaller households. Endemic areas typically coincide with regions of low socioeconomic resilience and limited healthcare access. This vulnerability, compounded by high-risk agricultural occupations, creates a significant public health challenge. However, ongoing urbanization and rural-urban migration may alter this pattern in the future.

## Unraveling the complex climatic and bioclimatic drivers of leptospirosis

Our analysis revealed complex relationships between climatic factors and leptospirosis risk. Through SHAP analysis, vapor pressure emerged as the most influential climatic variable, followed by maximum annual temperature, though these factors showed opposing effects. Surprisingly, the model assessed vapor pressure to negatively impact the leptospirosis risk, while precipitation (rainfall) is a positively impacting factor. Although vapor pressure (representing specific humidity) is a prerequisite for precipitation, their moderate correlation (Pearson's r = 0.3786, S2 Fig) suggests independent mechanisms of influence. While numerous studies have documented rainfall's role in leptospirosis risk across various countries [47–49], vapor pressure's influence has remained largely unexplored. Despite being the most significant climatic factor, vapor pressure ranked only fifth overall in our SHAP analysis, suggesting either a limited impact of annual climatic trends or a unique characteristic of leptospirosis epidemiology in Thailand. This finding aligns with previous studies reporting

modest climatic influences on annual leptospirosis risk [8,50]. Moreover, the timing of rainfall might be important to the leptospirosis outbreak, indicated by the precipitation during the driest month (*bio14*) as the 7th ranked factor.

### The impact of the COVID-19 pandemic on leptospirosis surveillance and risk patterns

Our results also highlight the impact of the COVID-19 pandemic on the leptospirosis outbreak in Thailand, one of the first countries outside China to detect COVID-19 [51,52]. Many control measures were implemented during 2020 and 2021, such as lockdowns, promoting social distancing, and wearing a facemask [52]. In the middle of the peak of the pandemic, Thailand's healthcare system is often overwhelmed with COVID-19 cases. This situation may be attributed to the underreporting of leptospirosis, as the non-severe cases may not seek hospital care. Additionally, underreporting can occur when healthcare systems are overwhelmed, making care unavailable for some patients. This underreporting and neglect of leptospirosis during the pandemic were also observed in other countries [53–56].

The deterioration in model performance during the COVID-19 pandemic is not unexpected, as our XGBoost model did not capture COVID-19-related interventions that likely disrupted leptospirosis transmission. This aligns with the typical performance decline observed in machine learning models when test set conditions differ from the training set [57,58]. Notably, SHAP importance rankings derived from pandemic-period data remained similar to those from the full dataset (S5 Fig), indicating that our model continued to rely on the same set of features to generate predictions despite the fundamentally altered epidemiological conditions during the COVID-19 pandemic. This consistency in feature importance, coupled with reduced predictive accuracy, suggests that the pandemic disrupted the established relationships between predictors and leptospirosis risk. Importantly, model performance recovered in 2022 following the lifting of pandemic control measures, achieving an AUC of 0.951 (95% CI: 0.861-0.999), demonstrating the model's robustness when applied to conditions similar to those in the training period.

The post-pandemic resurgence of cases in 2022 to the 2017 level—more than double the pandemic-era numbers—suggests significant leptospirosis surveillance disruption. Despite known mechanisms reported in other diseases, such as control measures disruption, reduced human movement, and surveillance distraction [59], the direct impact of COVID-19 disruption on leptospirosis remains to be uncovered, as the unique transmission pathway through the water of leptospirosis.

### Future directions and limitations

As the leptospirosis outbreak can impact the health of each individual and society at large, it is crucial to develop an early warning system to monitor and detect the beginning of the outbreak to limit the risk of exposure to individuals and properly manage the healthcare system. While leptospirosis risk emerges from complex interactions among environmental, climatic, and socioeconomic factors, the challenging nature of these relationships has historically limited the application of mechanistic and large-scale statistical modeling approaches. Our machine learning framework represents an initial step toward systematic risk factor identification and outbreak prediction, though several important limitations warrant consideration.

First, our analysis assumes consistent case reporting outside the pandemic period, potentially underestimating actual disease burden due to unreported mild and asymptomatic cases. Our data rely on a passive surveillance protocol, where under-reporting can occur non-randomly due to external factors. Reported cases from this passive surveillance system may be affected by unequal access to healthcare or limited diagnostic testing. Consequently, true incidence rates may differ from officially reported cases. Second, most of the data is available at a limited resolution, typically at the provincial and annual levels. Future research incorporating finer-scale data could significantly enhance predictive accuracy. Third, our binary risk classification, while effective, may oversimplify the complex risk landscape. A multiclass approach could provide more nuanced risk stratification and deeper insights into how climatic and socioeconomic factors influence disease patterns. Additionally, while our binary classification approach effectively distinguishes between high and low-risk provinces, this dichotomous categorization may mask essential gradations in risk levels. Machine learning algorithms

that directly predict continuous incidence rates, rather than binary outcomes, could provide more nuanced and actionable insights for stakeholders and authorities to support decision-making. Finally, our model and analysis identify feature associations rather than causal relationships with leptospirosis risk. More sophisticated analyses are needed to confirm causal relationships between these factors and leptospirosis risk.

Looking ahead, the burden of leptospirosis may intensify, particularly in economically disadvantaged regions, as climate change increases the frequency and severity of extreme weather events. However, our model's adaptable framework can readily incorporate new data sources and additional risk factors as they become available. This approach also offers a template for developing early warning systems for other infectious diseases at national scales, contributing to broader public health preparedness efforts.

## Supporting information

**S1 Text. XGBoost hyperparameters.**
(PDF)

**S1 Fig. Threshold sensitivity analysis of model performance on test data (2018–2019).** Performance metrics (accuracy, precision, sensitivity, and F1-score) are evaluated across different classification thresholds, where the threshold determines the cutoff between predicted low-risk (0) and high-risk (1) provinces. For each threshold value, provinces are classified as high-risk when their predicted probability exceeds the threshold.
(TIF)

**S2 Fig. Hierarchically clustered correlation matrix of predictive features.** The heatmap displays Pearson correlation coefficients between all model features, visualized using seaborn's clustermap function. Correlation strength and direction are indicated by color intensity (red = positive, blue = negative). Features are hierarchically clustered to reveal groups of highly correlated variables, highlighting potential multicollinearity in the dataset.
(TIF)

**S3 Fig. Temporal distribution of predicted high-risk provinces across pandemic periods.** Comparison of model-predicted versus observed counts of high-risk provinces during pre-pandemic (2007–2019), pandemic (2020–2021), and post-pandemic (2022) periods. Using the model trained on 2007–2017 data, predictions reveal substantial changes in the number of high-risk provinces, particularly during the pandemic period. Using a classification threshold of 0.5, the model predicted 36 and 34 high-risk provinces for 2020 and 2021, respectively, while actual data showed only 21 and 15 high-risk provinces. This overestimation suggests a substantial reduction in leptospirosis risk during the pandemic period, independent of climatic conditions. This visualization highlights the pandemic's potential impact on leptospirosis risk patterns and/or surveillance capabilities across Thailand's provinces.
(TIF)

**S4 Fig. Receiver operating characteristic (ROC) curves with area under the curve (AUC) values comparing model discrimination ability across three periods: pre-pandemic (2018–2019), pandemic (2020–2021), and post-pandemic (2022).** The diagonal reference line represents random classification (AUC = 0.5), while perfect classification corresponds to AUC = 1.0. Shaded areas indicate 95% confidence intervals based on 1,000 bootstrap iterations. 95% CIs of AUC are given in parentheses.
(TIF)

**S5 Fig. SHAP analysis of the top 10 predictive features, ranked by mean absolute SHAP values. (a)** Full dataset covering 2007–2022. **(b)** during COVID-19 pandemic data only (2020–2021).
(TIF)

**S1 Table. Performance comparison of four machine learning models (XGBoost, logistic regression, support vector machine, and random forest) using six metrics on training set, pre-COVID-19 (2018–2019) test set, and post-COVID-19 (2022) test set.** For each metric, ranking score calculated by summation of assigned rank from best (score = 4) to worst (score = 1) for each metric. The average performance of each model was obtained from average over 10 random seeds. All models were trained on the same training set (2007–2015) to ensure fair comparison. Hyperparameters were optimized using 5-fold cross-validation.
(PDF)

## Acknowledgments

We would like to thank our colleagues from the Mahidol-Bremen Medical Informatics Research Unit, Mahidol University, and Climate-sensitive Infectious Disease Lab, Heidelberg University, for the valuable discussion.

## Author contributions

**Conceptualization:** Sudarat Chadsuthi, Charin Modchang, Joacim Rocklöv.

**Data curation:** Pikkanet Suttirat, Sudarat Chadsuthi.

**Formal analysis:** Pikkanet Suttirat.

**Funding acquisition:** Charin Modchang.

**Investigation:** Pikkanet Suttirat.

**Methodology:** Pikkanet Suttirat, Sudarat Chadsuthi, Charin Modchang, Joacim Rocklöv.

**Project administration:** Charin Modchang, Joacim Rocklöv.

**Resources:** Pikkanet Suttirat, Sudarat Chadsuthi, Joacim Rocklöv.

**Software:** Pikkanet Suttirat.

**Supervision:** Sudarat Chadsuthi, Charin Modchang, Joacim Rocklöv.

**Validation:** Sudarat Chadsuthi, Charin Modchang, Joacim Rocklöv.

**Visualization:** Pikkanet Suttirat.

**Writing – original draft:** Pikkanet Suttirat, Charin Modchang.

**Writing – review & editing:** Sudarat Chadsuthi, Charin Modchang, Joacim Rocklöv.

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
