## [Decision Letter · Decision Letter 0]

17 Jul 2025

PNTD-D-25-00504

Unraveling the drivers of leptospirosis risk in Thailand using machine learning

Dear Dr. Modchang,

Thank you for submitting your manuscript to PLOS Neglected Tropical Diseases. After careful consideration, we feel that it has merit but does not fully meet PLOS Neglected Tropical Diseases's publication criteria as it currently stands. Therefore, we invite you to submit a revised version of the manuscript that addresses the points raised during the review process.

Please submit your revised manuscript within 60 days Sep 15 2025 11:59PM. If you will need more time than this to complete your revisions, please reply to this message or contact the journal office at plosntds@plos.org. Please include the following items when submitting your revised manuscript:

We look forward to receiving your revised manuscript.

Kind regards,

Mathieu Picardeau

Section Editor

Shaden Kamhawi

co-Editor-in-Chief

Paul Brindley

co-Editor-in-Chief

**Journal Requirements:**

3) Some material included in your submission may be copyrighted. According to PLOSu2019s copyright policy, authors who use figures or other material (e.g., graphics, clipart, maps) from another author or copyright holder must demonstrate or obtain permission to publish this material under the Creative Commons Attribution 4.0 International (CC BY 4.0) License used by PLOS journals. Please closely review the details of PLOSu2019s copyright requirements here: PLOS Licenses and Copyright. If you need to request permissions from a copyright holder, you may use PLOS's Copyright Content Permission form.

Potential Copyright Issues:

- Figure 1B and 1C. Please (a) provide a direct link to the base layer of the map (i.e., the country or region border shape) and ensure this is also included in the figure legend; and (b) provide a link to the terms of use / license information for the base layer image or shapefile. We cannot publish proprietary or copyrighted maps (e.g. Google Maps, Mapquest) and the terms of use for your map base layer must be compatible with our CC BY 4.0 license.

4) Please ensure that the funders and grant numbers match between the Financial Disclosure field and the Funding Information tab in your submission form. Note that the funders must be provided in the same order in both places as well. State the initials, alongside each funding source, of each author to receive each grant. For example: "This work was supported by the National Institutes of Health (####### to AM; ###### to CJ) and the National Science Foundation (###### to AM).".

**Reviewers' Comments:**

Reviewer's Responses to Questions

**Key Review Criteria Required for Acceptance?**

**Methods**

-Are the objectives of the study clearly articulated with a clear testable hypothesis stated?

-Is the study design appropriate to address the stated objectives?

-Is the population clearly described and appropriate for the hypothesis being tested?

-Is the sample size sufficient to ensure adequate power to address the hypothesis being tested?

-Were correct statistical analysis used to support conclusions?

-Are there concerns about ethical or regulatory requirements being met?

Reviewer #1: Are the objectives of the study clearly articulated with a clear testable hypothesis stated?Yes

-Is the study design appropriate to address the stated objectives? Yes

-Is the population clearly described and appropriate for the hypothesis being tested?Yes

-Is the sample size sufficient to ensure adequate power to address the hypothesis being tested?Yes

-Were correct statistical analysis used to support conclusions? Yes

-Are there concerns about ethical or regulatory requirements being met? No concerns

Reviewer #2: I am not qualified to critique application of the AI model.

A key limitation of the method is the reliance on passive surveillance for identifying cases which will undoubtably under-report cases in a non-random way. In some way the authors acknowledge this with their interpretation of the 2020-2021 COVID-19 affected data. While I appreciate that a two decade active surveillance programme is not feasible, it would be good to explore this limitation.

I was a bit concerned that when the model performed less well (post-2019) the data was excluded from analysis. Could this indicate an inability to generalise your findings?

**Results**

-Does the analysis presented match the analysis plan?

-Are the results clearly and completely presented?

-Are the figures (Tables, Images) of sufficient quality for clarity?

Reviewer #1: Yes the results and analysis is clear with the figure and tables of sufficient quality.

Reviewer #2: Is it possible to place confidence intervals around your point estimates of AUC? I think these are important in considering the precision of your estimates.

**Conclusions**

-Are the conclusions supported by the data presented?

-Are the limitations of analysis clearly described?

-Do the authors discuss how these data can be helpful to advance our understanding of the topic under study?

-Is public health relevance addressed?

Reviewer #1: Yes the conclusions are clear with limitations clearly described in the last section. Yes the public health relevance is addressed.

Reviewer #2: It would be very helpful to explore the limitations of your methods - for example, challenges with variation in reported cases might reflect variable barriers to presentation or diagnostic testing as well as variation in incidence.

Your model reflects association rather than (neccessarily) causation, and it would be useful for readers to highlight this more.

**Editorial and Data Presentation Modifications?**

Reviewer #1: There is a small typo in the code on line 49 (the name of the meta data file).

Reviewer #2: (No Response)

**Summary and General Comments**

Reviewer #1: The paper is nicely formatted and clear on the goals and the overall results.

However, it would be nice to see an extra experiment done where a more basic model is fit to justify the usage of a more complex model such as xgboost. This should be relatively straight forward experiment to run.

Another point which I wouldn't mind addressed either as future work/limitation or another experiment is how well the model can fit the incidence rate. The usefulness of this is that it would allow a user/stakeholder to decide what high to low risk means.

When describing the problem around line 179 it may be best to be more exact (it is described exactly in the results)

Finally, although the code included covers the diagrams it would be nice if the complete code could be included i.e. hyperparameter tuning. e 49)

Reviewer #2: (No Response)

PLOS authors have the option to publish the peer review history of their article (what does this mean? ). If published, this will include your full peer review and any attached files.

**Do you want your identity to be public for this peer review?** For information about this choice, including consent withdrawal, please see our Privacy Policy .

Reviewer #1: No

Reviewer #2: No

**Figure resubmission:**
---

## [Decision Letter · Decision Letter 1]

3 Oct 2025

Dear Dr. Modchang,

We are pleased to inform you that your manuscript 'Unraveling the drivers of leptospirosis risk in Thailand using machine learning' has been provisionally accepted for publication in PLOS Neglected Tropical Diseases.

Best regards,

Mathieu Picardeau

Section Editor

Shaden Kamhawi

co-Editor-in-Chief

Paul Brindley

co-Editor-in-Chief

Reviewer's Responses to Questions

**Key Review Criteria Required for Acceptance?**

**Methods**

-Are the objectives of the study clearly articulated with a clear testable hypothesis stated?

-Is the study design appropriate to address the stated objectives?

-Is the population clearly described and appropriate for the hypothesis being tested?

-Is the sample size sufficient to ensure adequate power to address the hypothesis being tested?

-Were correct statistical analysis used to support conclusions?

-Are there concerns about ethical or regulatory requirements being met?

Reviewer #1: Are the objectives of the study clearly articulated with a clear testable hypothesis stated?Yes

-Is the study design appropriate to address the stated objectives? Yes

-Is the population clearly described and appropriate for the hypothesis being tested?Yes

-Is the sample size sufficient to ensure adequate power to address the hypothesis being tested?Yes

-Were correct statistical analysis used to support conclusions? Yes

-Are there concerns about ethical or regulatory requirements being met? No concerns

Reviewer #2: (No Response)

**Results**

-Does the analysis presented match the analysis plan?

-Are the results clearly and completely presented?

-Are the figures (Tables, Images) of sufficient quality for clarity?

Reviewer #1: Yes the results and analysis is clear with the figure and tables of sufficient quality.

Reviewer #2: (No Response)

**Conclusions**

-Are the conclusions supported by the data presented?

-Are the limitations of analysis clearly described?

-Do the authors discuss how these data can be helpful to advance our understanding of the topic under study?

-Is public health relevance addressed?

Reviewer #1: Yes the conclusions are clear with limitations clearly described in the last section. Yes the public health relevance is addressed.

Reviewer #2: (No Response)

**Editorial and Data Presentation Modifications?**

Reviewer #1: (No Response)

Reviewer #2: (No Response)

**Summary and General Comments**

Reviewer #1: The authors have addressed my previous comments.

Reviewer #2: The authors have addressed my previous comments. I have no additional critique of the paper.

PLOS authors have the option to publish the peer review history of their article (what does this mean? ). If published, this will include your full peer review and any attached files.

**Do you want your identity to be public for this peer review?** For information about this choice, including consent withdrawal, please see our Privacy Policy .

Reviewer #1: No

Reviewer #2: No

---

## [Editor Report · Acceptance letter]

Dear Dr. Modchang,

We are delighted to inform you that your manuscript, "Unraveling the drivers of leptospirosis risk in Thailand using machine learning," has been formally accepted for publication in PLOS Neglected Tropical Diseases.

Best regards,

Shaden Kamhawi

co-Editor-in-Chief

Paul Brindley

co-Editor-in-Chief
